# Analysis of Landscape Pattern Evolution and Driving Forces Based on Land-Use Changes: A Case Study of Yilong Lake Watershed on Yunnan-Guizhou Plateau

**Guoqiang Ma** [1,2], **Qiujie Li** [3], **Shuyu Yang** [1], **Rong Zhang** [1], **Lixun Zhang** [2], **Jianping Xiao** [1] **and Guojun Sun** [2,*]

1   Southwest Investigation and Planning Institute of State Forestry and Grassland Administration, Kunming 650216, China
2   College of Ecology, Lanzhou University, Lanzhou 730000, China
3   Power China Kunming Engineering Corporation Limited, Kunming 650051, China
*   Correspondence: sungj@lzu.edu.cn

**Abstract:** In order to explore the landscape pattern evolution and driving forces of the Yilong Lake watershed, the combined method of supervised classification with manual visual interpretation based on the landsat5TM/8OLI remote sensing image data sources was used to establish a high-precision spatial distribution information database of the Yilong Lake watershed. Landscape index was used to analyze the distribution and spatial pattern change characteristics of various land-use types. Based on correlation and principal component analysis, we discuss the relationship between the change characteristics of land-use type, distribution and spatial pattern, and the interference of local socio-economic development and natural factors. The results show that: (1) In the past 30 years, the land-use types of the Yilong Lake watershed are mainly forest, garden plot and cultivated land. The forest area decreased significantly by 30.45 km$^2$, of which the fastest reduction stage was from 2000 to 2005, with a total reduction of 20.56 km$^2$. The garden plot conversion is relatively large, with a total of 181.69 km$^2$ transferred out, of which 28.84 km$^2$ has become unused land, respectively. (2) In the past 30 years, the maximum patch index decreased by 9.94% and the patch density index increased by 14.25%, indicating that the landscape fragmentation in the whole basin increased. The Shannon diversity index showed an increasing process; the aggregation index showed a decreasing process. (3) The change in landscape pattern in the watershed was closely related to economic growth, population growth, social affluence and agricultural development. Natural factors, social factors and economic indicators are significantly positively correlated with patch density, edge density, landscape shape index and Shannon diversity index, and significantly negatively correlated with the largest patch index and the contagion index. On the whole, the wetlands in the basin are shrinking and the landscape diversity is changing. Reducing the excessive impact of human activities on the watershed ecosystem is a key factor for the local protection of wetland resources and the maintenance of wetland ecological functions.

**Keywords:** Yilong Lake watershed; landscape pattern; land use; driving factors

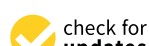



## 1. Introduction

Plateau lake wetlands are an important part of China's "two screens and three belts" ecological security pattern, and play an irreplaceable role in water conservation, biodiversity protection, and carbon fixation [1]. Yilong Lake is a plateau freshwater lake at the southernmost tip of China, an important channel for migratory birds to migrate, a typical wetland resource type in the Yunnan Guizhou Plateau, and one of the nine lakes in Yunnan Province [2]. Compared with other plateau lakes in Yunnan Province, Yilong Lake has a special history and has undergone many large-scale transformations [3,4]. In recent years, due to the dual constraints of the natural environment and human production and life, the wetland ecosystem is fragile and sensitive, and the wetland structure and ecological

function degradation are obvious [5]. The wetland ecosystem is an important condition for maintaining regional social, economic, and environmentally sustainable development [6]. In recent decades, due to the influence of climate and environmental factors and human activities [7–9], the water level of plateau lakes has decreased, the area has decreased [10], the swamp wetland has shrunk and degraded [11], and the area, type, structure, and function of wetlands have changed significantly, especially the change inland use. Land is the most basic natural resource and the material basis of human economic and social activities. Land use/land cover (LULC) change is a direct characterization signal of various human production activities acting on the ecological environment on the earth's surface [12,13]. The diversity of spatial composition and distribution of land use indicates the spatial distribution characteristics of various types of wetland patches [14]. The temporal and spatial evolution of landscape pattern is the most intuitive form of land-use change [15]. The analysis of landscape pattern evolution based on land-use change is an important content of landscape ecology research. It has played a role in promoting the study of plateau lake wetland landscape on a large scale.

It is very important to analyze the driving mechanism of landscape pattern, which can help find out the causes of landscape pattern change [16]. Natural factors dominated by climate change and human activities represented by social and economic development are the main driving factors leading to the continuous reduction in wetland distribution areas [17,18]. In recent years, many scholars have carried out a great deal of research on the wetland evolution process and mechanism by using 3 S technology and landscape ecology theory [19]. The existing research on landscape pattern usually selects landscape-pattern-related indexes such as area, shape, aggregation and diversity to measure and reflect the characteristics and composition of landscape patches. This study examines the role and contribution of natural factor changes and human activity interference in the evolution of the wetland ecosystem, based on qualitative analysis, correlation analysis and regression analysis [20,21]. There are fewer amounts of data used in correlation analysis, which can be better applied to the needs of plateau lake wetland analysis. In view of this, taking the Yilong Lake Basin on the Yunnan Guizhou Plateau as the research object, through the research on the land-use change, landscape pattern evolution and the correlation with natural factors and human factors in the Yilong Lake Basin in the past 30 years, this paper explores the driving factors of landscape pattern evolution under the land-use change in the Yilong Lake Basin. It provides a scientific basis for future sustainable development, ecological environment protection and regional management of the Yilong Lake Basin.

## 2. Materials and Methods

### 2.1. Study Area

Shiping County is located in the southeast of the Yunnan Province and in the west of Honghe Hani and Yi Autonomous Prefecture. It spans between longitude 102°8′–102°43′ E and latitude 23°19′–24°6′ N. Yilong Lake is the southernmost plateau freshwater lake in China. It is a typical wetland resource type in the Yunnan Guizhou Plateau and one of the nine largest lakes in Yunnan Province. The Yilong Lake watershed mainly includes Yilong Town, Baoxiu town and Bazin town of Shiping County, in an east–west strip. The drainage area is 357.61 km$^2$ (Figure 1).

### 2.2. Data Source and Processing

The remote sensing image data of the Yilong Lake Basin are landsat5 TM (1990, 1995, 2000, 2005, 2010), landsat8 oli (2015, 2018) Landsat series data. The base map coordinate system is (GCS\china\geodetic\coordinate\system\2000), and the spatial resolution is 30 m. The RGB band synthesis and spatial geographic correction are carried out on the image. Land-use types in the basin are divided according to the land resource classification system [22] and natural language classification [23]. Natural language classification is a process of considering various functions of land in the natural environment and human social development, and dividing it into several functional areas [24]. Among the various

functions of the natural language system, ecological function is the premise and foundation of other functions [25]. At the same time, combined with field investigations, on-site visits and the collection of the latest data, a remote sensing interpretation mark is established after comprehensive analysis. The classification system includes three main categories and seven secondary categories (Table 1, Figure 2).

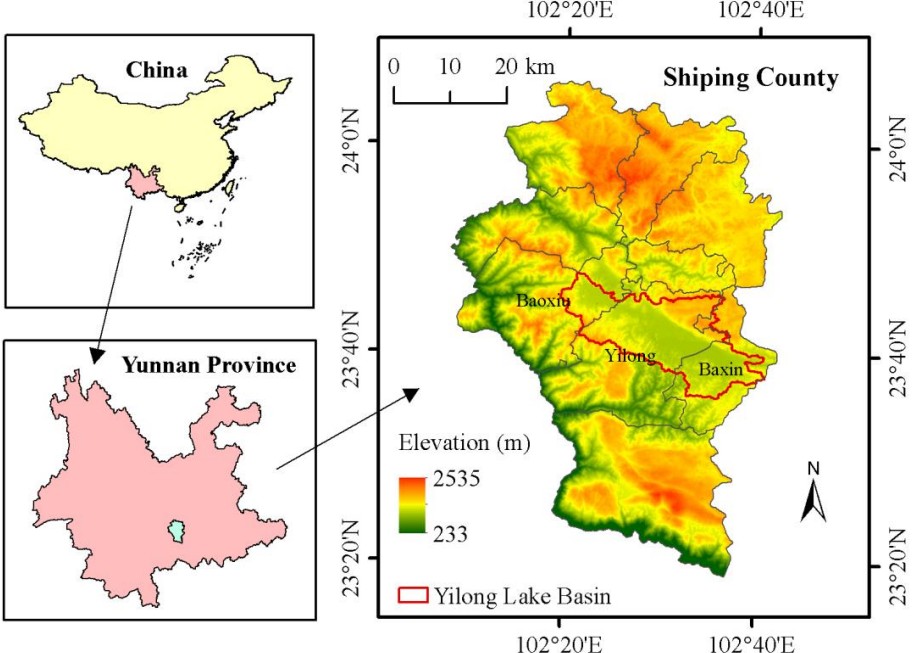

**Figure 1.** The map of Yilong Lake watershed.

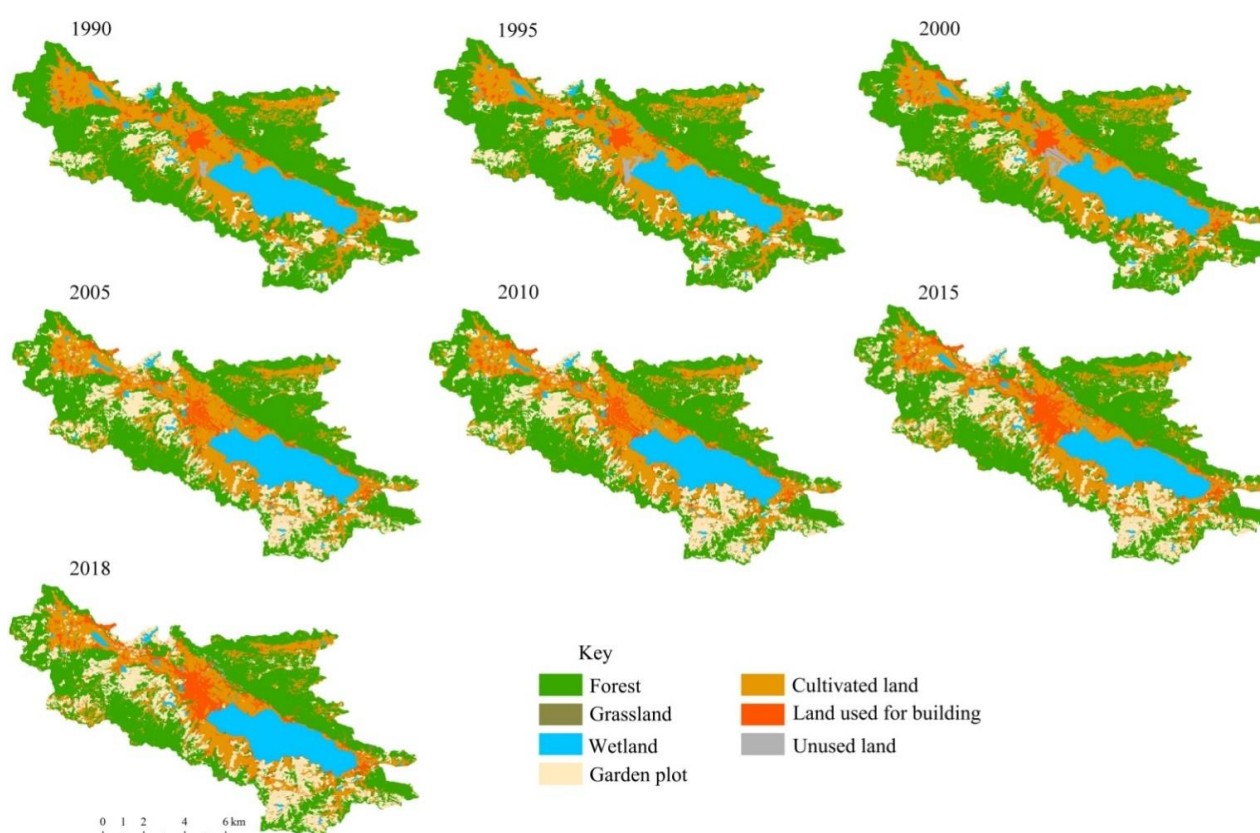

**Figure 2.** Spatial variation of landscape types in time series.

**Table 1.** Classification of land-use system in the Yilong Lake Basin.

| Main Categories | Secondary Category | Corresponding Land-Use Types |
|---|---|---|
| Ecological space | Forest | Bamboo forest land, arbor forest land, shrub land and other forest lands |
| | Grassland | grassland |
| | Wetland | Lakes, river wetlands, swamp wetlands, artificial wetlands |
| Production space | Garden plot | Garden plot |
| | Cultivated land | Paddy field, dry land |
| Living space | Land used for building | Construction land for urban and rural residents, industrial and mining construction land, and transportation land |
| | Unused land | Bare ground |

*2.3. Method*

2.3.1. Land-Use Change

(1) Dynamic degree of land use; it refers to the quantitative change inland-use types within a certain time range in the study area, reflecting the impact of human activities on land-use types [26]. By quantitatively describing the change speed of land-use types, its development trend can be predicted. The expression of dynamic degree of land-use type is:

$$K = \frac{U_b - U_a}{U_a} \times \frac{1}{T} \times 100\%$$

where: $K$ is the dynamic degree of a certain landscape type in a certain period of time; $U_a$ and $U_b$ are the area of a certain landscape type at the initial stage and the final stage of the study, hm$^2$; $T$ is the time, *a*.

(2) Land-use transfer matrix; it contains the static data of land-use types in a certain period of time and the dynamic data of mutual transformation of various land types. The transfer direction and quantity of different types of land use are analyzed through the transfer matrix, and the general form of land-use transfer matrix refers to the relevant literature [27]. This study uses the calculation method to analyze the internal transfer of land-use types in the study area.

$$S_{ij} = \begin{bmatrix} S_{11} & \cdots & \cdots S_{1n} \\ \cdots & \cdots & \cdots \cdots \\ S_{n1} & \cdots & \cdots S_{nn} \end{bmatrix}$$

where $S$ represents the area, $n$ represents the number of land-use types before and after the transfer, $i$ represents the land-use type before the transfer, and $j$ represents the land-use type after the transfer, which is the area from land class $i$ to land class $j$. Each row of elements in the matrix represents the local class source information of each local class before the transfer of land class $j$.

2.3.2. Landscape Index Analysis

Based on the correlation between landscape indexes [28,29], the landscape indexes of landscape pattern evolution in the Yilong Lake watershed were screened and analyzed from three aspects: landscape fragmentation, landscape shape and landscape diversity [30]. From the category level, select the largest patch index (LPI), patch density (PD), edge density (ED) and landscape shape index (LSI); from the landscape level, select the largest patch index (LPI), patch density (PD), edge density (ED), landscape shape index (LSI), the contagion index (CONTAG) and Shannon diversity index (SHDI). FRAGSTATS 4.2 (Home Page, China) software was used to analyze the changes in landscape pattern from three aspects: landscape fragmentation (LPI and PD), landscape patch shape (ED and LSI) and landscape diversity (CONTAG and SHDI).

2.3.3. Collection and Treatment of Human and Natural Factors

Regional socio-economic development indicators can comprehensively reflect the type and intensity of the interference of human activities on natural resources [31]. The watershed is a typical agricultural watershed. Select the total population (TP), urban population (UP), rural population (RP), agriculture, forestry, animal husbandry and fishery industry (AF), the forestry output value (F), the total sown area of crops (TC), the large livestock breeding (LLB) and the pig breeding (PB) as social driving indicators, gross domestic product (GDP), the primary industry (PI), the second industry (SI), and the third industry (TI) as economic driving indicators. The annual average precipitation (P) (mm) and the annual mean temperature (T) (°C) are used as natural driving indicators. Other data are from the statistical yearbook of Shiping County (1990–2018).

SPSS 22 was used to analyze the correlation between the regional landscape index and the output value of social and economic development, as well as natural factors. Principal component analysis was used to analyze the impact and contribution of human activity interference and natural factor changes on the change in regional landscape diversity.

## 3. Results

### 3.1. Dynamic Change Characteristics of Land Use

According to Figure 2 and Table 2, the land-use types of the Yilong Lake Basin in the last 30 years are mainly forests, gardens and cultivated land. In the last30 years, the forest area has decreased significantly by 30.45 km$^2$, of which the fastest reduction stage is from 2000 to 2005, with a total reduction of 20.56 km$^2$ and a dynamic degree of ("−" U+2.31%). At the same time, the garden area has increased significantly, with an increase of 32.53 km$^2$ in the past 30 years. Among them, the fastest period of increase was from 2000 to 2005, with a total increase of 27.21 km$^2$ and a dynamic degree of 16.23%. The wetland area fluctuates greatly, first increasing, then decreasing, and then increasing. From 1990 to 2015, the lake area decreased by 3.83 km$^2$, and in 2018, the lake area was 40.42 km$^2$, and from 2015 to 2018, the lake area increased by 2.46 km$^2$. In the past 30 years, the construction land and transportation land for urban and rural residents have increased rapidly. The fastest period of area increase is from 2000 to 2015. The construction land for urban and rural residents has increased by 17.19 km$^2$, with a dynamic degree of 7.38%.

**Table 2.** The changes in land-use area in the Yilong Lake watershed from 1990 to 2018.

| Year | Index | Forest | Grassland | Wetland | Garden Plot | Cultivated Land | Land Used for Building | Unused Land |
|------|-------|--------|-----------|---------|-------------|-----------------|------------------------|-------------|
| 1990 | | 181.66 | 41.79 | 0.15 | 30.02 | 89.34 | 13.84 | 0.83 |
| 1995 | | 178.84 | 41.62 | 0.25 | 33.28 | 87.41 | 14.27 | 1.95 |
| 2000 | | 178.32 | 42.06 | 0.25 | 33.53 | 85.01 | 15.53 | 2.9 |
| 2005 | Area/km$^2$ | 157.76 | 40.04 | 0.58 | 60.75 | 76.49 | 21.56 | 0.41 |
| 2010 | | 154.06 | 39.91 | 0.7 | 62.08 | 76.38 | 24.04 | 0.43 |
| 2015 | | 151.28 | 37.96 | 0.91 | 62.56 | 71.22 | 32.72 | 0.94 |
| 2018 | | 151.21 | 40.42 | 0.93 | 62.54 | 69.17 | 32.4 | 0.94 |
| 1990–1995 | Area/km$^2$ | −2.82 | −0.17 | 0.1 | 3.26 | −1.93 | 0.43 | 1.12 |
| | Dynamic degree/% | −0.31 | −0.08 | 13.33 | 2.17 | −0.43 | 0.62 | 26.99 |
| 1995–2000 | Area/km$^2$ | −0.52 | 0.44 | 0 | 0.25 | −2.4 | 1.26 | 0.95 |
| | Dynamic degree/% | −0.06 | 0.21 | 0.00 | 0.15 | −0.55 | 1.77 | 9.74 |
| 2000–2005 | Area/km$^2$ | −20.56 | −2.02 | 0.33 | 27.22 | −8.52 | 6.03 | −2.49 |
| | Dynamic degree/% | −2.31 | −0.96 | 26.40 | 16.24 | −2.00 | 7.77 | −17.17 |
| 2005–2010 | Area/km$^2$ | −3.7 | −0.13 | 0.12 | 1.33 | −0.11 | 2.48 | 0.02 |
| | Dynamic degree/% | −0.47 | −0.06 | 4.14 | 0.44 | −0.03 | 2.30 | 0.98 |
| 2010–2015 | Area/km$^2$ | −2.78 | −1.95 | 0.21 | 0.48 | −5.16 | 8.68 | 0.51 |
| | Dynamic degree/% | −0.36 | −0.98 | 6.00 | 0.15 | −1.35 | 7.22 | 23.72 |
| 2015–2018 | Area/km$^2$ | −0.07 | 2.46 | 0.02 | −0.02 | −2.05 | −0.32 | 0 |
| | Dynamic degree/% | −0.02 | 2.16 | 0.73 | −0.01 | −0.96 | −0.33 | 0.00 |
| 1990–2018 | Area/km$^2$ | −30.45 | −1.37 | 0.78 | 32.52 | −20.17 | 18.56 | 0.11 |
| | Dynamic degree/% | −0.60 | −0.12 | 18.57 | 3.87 | −0.81 | 4.79 | 0.47 |

It can be seen from Table 3 that from 1990 to 2018, the garden land was the most frequently converted in the Yilong Lake Basin, with a total of 181.69 km$^2$, of which 28.84 km$^2$ were turned to unused land, respectively. The second is grassland, with a total area of 89.34 km$^2$, mainly turning to wetlands, with a total area of 13.42 km$^2$. A total of 41.77 km$^2$ of cultivated land has been transferred out, and the main grassland is 2.19 km$^2$.

**Table 3.** The conversion of land-use types in the Yilong Lake watershed from 1990 to 2018 (Unit: km$^2$).

| 1990–2018 | Forest | Grassland | Wetland | Garden Plot | Cultivated Land | Land Used for Building | Unused Land | Total |
|---|---|---|---|---|---|---|---|---|
| Forest | 0.11 | 0.00 | | 0.01 | | | 0.02 | 0.14 |
| Grassland | 0.01 | 57.00 | 13.42 | 6.98 | 1.64 | 0.18 | 10.11 | 89.34 |
| Wetland | 0.01 | 1.05 | 11.55 | 0.75 | 0.01 | 0.11 | 0.37 | 13.86 |
| Garden plot | 0.77 | 6.39 | 4.81 | 140.25 | 0.11 | 0.53 | 28.84 | 181.69 |
| Cultivated land | 0.01 | 2.19 | 0.63 | 0.12 | 38.57 | 0.09 | 0.16 | 41.77 |
| Land used for building | | 0.02 | 0.78 | | | | 0.02 | 0.83 |
| Unused land | 0.03 | 2.45 | 1.27 | 3.09 | 0.11 | 0.03 | 23.00 | 29.97 |
| Total | 0.93 | 69.11 | 32.47 | 151.21 | 40.43 | 0.94 | 62.52 | 357.61 |

### 3.2. Change in Landscape Pattern Index

As shown in Figure 3, from 1990 to 2018, the maximum patch index at the landscape level decreased by 9.94%, and the patch density index increased by 14.25%, indicating that the landscape fragmentation in the whole basin increased. At the level of land-use types, the patch density index of all land-use types has shown an upward trend in the past 30 years, with the most drastic change in construction land, increasing from 0.62 to 11.29. The largest patch index in the basin increased with construction land and garden land, with construction land increasing from 0.91 to 3.32 and garden land increasing from 0.98 to 3.60. Forest and cultivated land decreased the most, with forest decreasing from 25.85 to 15.91 and cultivated land decreasing from 15.26 to 3.83. In the past 30 years, the patch density index of construction land has increased more than the maximum patch index, indicating that construction land is mainly expanding away from the urban area.

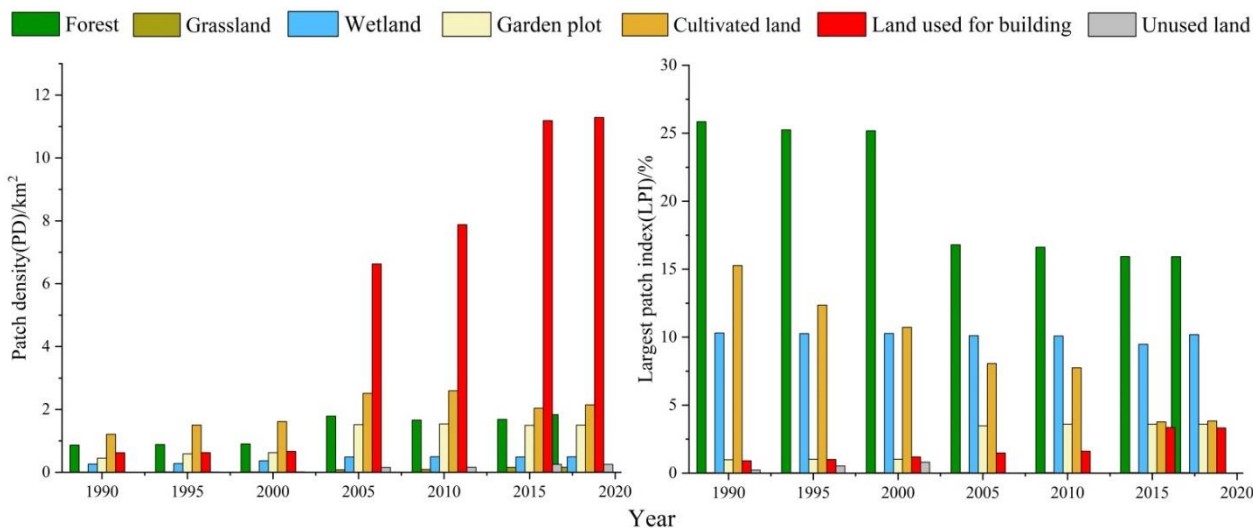

**Figure 3.** Changes in landscape fragmentation in the Yilong Lake watershed.

According to Figure 4, in the past 30 years, the landscape shape index and edge density index have shown a growing trend at the landscape level, indicating that the shape of landscape patches in the basin is becoming increasingly complex. The landscape shape index of all land-use types showed different degrees of growth, including construction land

increased by 27.35, garden land and forest increased by 21.48 and 13.11, respectively. Except for the decrease in the wetland edge density index, the edge density index of other land types increased, and the edge density index of construction land increased the most, by 34.28. It can be seen that the changes in the landscape patch shape in the basin are mainly affected by the changes in urban and rural residents' construction land, transportation land and cultivated land.

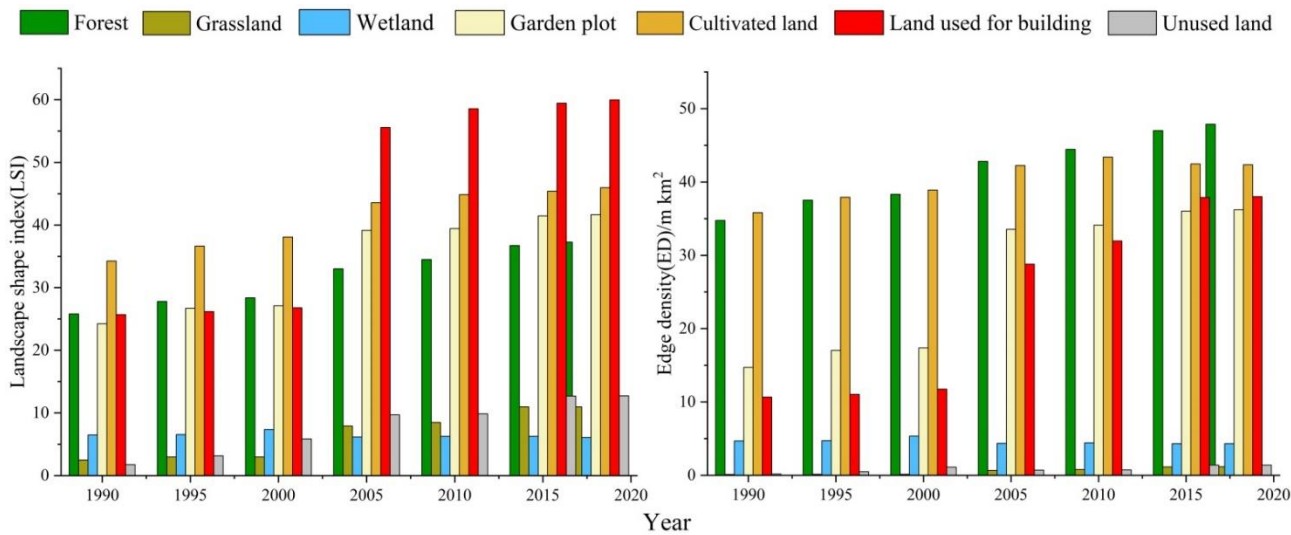

**Figure 4.** Change in landscape patch shape in the Yilong Lake watershed.

It can be seen from Table 4 that in the past 30 years, the landscape diversity in the basin has increased, the aggregation index has decreased by 9.56, and the Shannon evenness index has increased by 0.19, but the change intensity is lower than the landscape fragmentation and the change in landscape patch shape. In the past 30 years, the variation range of landscape diversity index in the basin is small. The change center of the landscape diversity index is around the construction land of urban and rural residents, and the surface of the Yilong Lake. Shannon diversity index showed an increasing process, indicating that the richness of the landscape increased during this period. The aggregation index showed a decreasing process, indicating that during this period, the connectivity between landscape types increased and the overall heterogeneity decreased. The water landscape shows the most significance when the dominance and connectivity of the landscape types increase.

**Table 4.** Landscape pattern index of the Yilong Lake watershed 1990–2018.

| Year | PD (/km$^2$) | LPI (%) | LSI | ED | CONTAG (%) | SHDI |
|------|------|------|------|------|------|------|
| 1990 | 3.4227 | 25.85 | 26.1209 | 50.4526 | 58.2824 | 1.2925 |
| 1995 | 3.8953 | 25.2454 | 27.9968 | 54.4215 | 56.9152 | 1.3242 |
| 2000 | 4.1945 | 25.183 | 28.9619 | 56.4634 | 56.1037 | 1.3426 |
| 2005 | 13.1652 | 16.79 | 38.4485 | 76.5342 | 51.46 | 1.4248 |
| 2010 | 14.4124 | 16.6096 | 40.0508 | 79.9242 | 50.5428 | 1.4431 |
| 2015 | 17.2996 | 15.9195 | 42.5397 | 85.0927 | 48.867 | 1.478 |
| 2018 | 17.6757 | 15.9127 | 42.755 | 85.6456 | 48.7175 | 1.4818 |

*3.3. Relationship between Watershed Landscape Change and Natural, Social and Economic Development*

According to Table 5, Principal component analysis showed that the influence weight of social key factors on wetland area and landscape diversity was greater than that of the annual average temperature and precipitation. The results showed that the explanatory degrees of the first and second principal components to the total variables were 78.03% and 17.07%, respectively. Among them, human activity factors, namely, the output value of

agriculture, forestry, animal husbandry and fishery, the value of the tertiary industry, the value of the secondary industry and the value of the primary industry, have a large load on the first principal component. These indicators mainly reflect the status of social and economic development. Therefore, it can be considered that the first principal component is the representative of social economy. The annual average temperature and precipitation among the natural driving factors have a large load on the second principal component; thus, the second principal component is considered to be representative of climate factors.

**Table 5.** Correlation between the landscape index and key social and natural factors in the study area.

| Drivers | Landscape Index | | | | | |
|---------|----------|----------|----------|----------|----------|----------|
| | **PD** | **LPI** | **LSI** | **ED** | **CONTAG** | **SHDI** |
| TP | 0.963 ** | −0.923 ** | 0.970 ** | 0.970 ** | −0.980 ** | 0.984 ** |
| RP | −0.238 | 0.091 | −0.186 | −0.186 | 0.211 | −0.224 |
| UP | 0.784 * | −0.670 | 0.757 * | 0.758 * | −0.779 * | 0.789 * |
| T | 0.935 ** | −0.908 ** | 0.937 ** | 0.937 ** | −0.935 ** | 0.934 ** |
| P | −0.143 | 0.235 | −0.184 | −0.185 | 0.170 | −0.162 |
| PI | 0.883 ** | −0.789 * | 0.866 * | 0.867 * | −0.884 ** | 0.891 ** |
| SI | 0.780 * | −0.669 | 0.752 | 0.752 | −0.771 * | 0.779 * |
| TI | 0.795 * | −0.683 | 0.767 * | 0.768 * | −0.786 * | 0.795 * |
| AF | 0.907 ** | −0.819 * | 0.888 ** | 0.888 ** | −0.900 ** | 0.905 ** |
| F | 0.922 ** | −0.843 * | 0.901 ** | 0.901 ** | −0.911 ** | 0.914 ** |
| TC | 0.961 ** | −0.917 ** | 0.966 ** | 0.966 ** | −0.976 ** | 0.980 ** |
| LLB | 0.820 * | −0.723 | 0.802 * | 0.803 * | −0.819 * | 0.826 * |
| PB | 0.754 | −0.696 | 0.750 | 0.751 | −0.757 * | 0.758 * |

Note: * $p < 0.05$, ** $p < 0.01$.

It can be seen from Table 4 that, on the whole, the selected social key factors have a strong correlation with the six landscape indexes, indicating that the change in the landscape pattern in the Yilong Lake watershed is closely related to the human factors and social and economic development in the watershed. Based on the analysis of three landscape directions, the fragmentation degree: patch density and largest patch index. Patch density was positively correlated with population, urban population, annual average temperature, value of primary, secondary and tertiary industries, agriculture, forestry, animal husbandry and fishery, forestry output value, total sown area of crops and large livestock breeding. It has a negative correlation with the rural population and precipitation, and the correlation is not significant. Largest patch index is negatively correlated with population, annual average temperature, the value of primary and tertiary industries, agriculture, forestry, animal husbandry and fishery, forestry output value, total sown area of crops, large livestock breeding and pig breeding.

Shape: Edge density and landscape shape index. Edge density was positively correlated with population, urban population, annual average temperature, value of primary, secondary and tertiary industries, agriculture, forestry, animal husbandry and fishery, forestry output value, total sown area of crops, large livestock breeding and pig breeding. It has a negative correlation with the rural population and precipitation, and the correlation is not significant. The landscape shape index is significantly positively correlated with population, urban population, annual average temperature, value of primary, secondary and tertiary industries, agriculture, forestry, animal husbandry and fishery, forestry output value, total sown area of crops, large livestock breeding and pig breeding. It has a negative correlation with the rural population and precipitation, and the correlation is not significant.

Diversity: The contagion index and Shannon diversity index. The contagion index is negatively correlated with population, urban population, annual average temperature, value of primary, secondary and tertiary industries, agriculture, forestry, animal husbandry and fishery, forestry output value, total sown area of crops and large livestock breeding. It is positively correlated with the rural population, and the correlation is not significant. The

Shannon diversity index was positively correlated with population, the value of primary, secondary and tertiary industries, agriculture, forestry, animal husbandry and fishery, forestry output value, total sown area of crops and large livestock breeding. It is negatively correlated with the rural population, and the correlation is not significant.

## 4. Discussion

### 4.1. Dynamic Change Analysis of Land Use

In this study, a high-precision spatial distribution information database of the Yilong Lake Basin was established based on seven phases of landsat5TM/8OLI remote sensing image data sources from 1990 to 2018 by combining supervised classification and manual visual interpretation, and 15 land-use types were extracted. Our research shows that the land use in the Yilong Lake Basin has changed dramatically in the past 30 years, which is mainly manifested in the reduction in forest and cultivated land area, the substantial increase in construction land, and the fragmentation of the landscape pattern, the complexity of the shape, and the increase in landscape diversity. The increase in the area of construction land has led to a significant reduction in the area of cultivated land. From 1990 to 2018, the total area of cultivated land has decreased by 20.17 km$^2$, a decrease of about 22.58%. Among them, the fastest reduction period is from 2000 to 2015, a total decrease of 13.79 km$^2$. Generally, the interference of human activities on wetlands mainly acts on small spatial and temporal scales, and has the characteristics of diverse methods, high intensity, fast speed, relatively short duration, etc. [32]. The landscape shape index and edge density index are both showing a growing trend, indicating that the shape of landscape patches in the watershed is becoming increasingly complex. In the past 30 years, there has been a frequent process of mutual transformation among various land-use types in the Yilong Lake Basin, with the continuous reduction in cultivated land and a large number converted into garden land and construction land. The cultivated land in the Yilong Lake Basin continues to decrease, which is the fastest among all landscape types. The change trend of forest and construction land in these periods shows a stable growth, of which the construction land is the most stable relative to the growth. The frequent transformation and changes between these lands are related to the specific policy environment at that time, such as the transformation policy of medium- and low-yield forests and the change in regional real estate market policy. While human influence acts on the landscape pattern, the evolution of the landscape pattern will also affect human production and social activities in the process of evolution [33]. From 1995 to 2015, the lake area decreased by 14.64 km$^2$, mainly due to the impact of human activities, social and economic development has a stronger impact on the land-use pattern, and the water level of the Yilong Lake continues to decline [4,34]. At this time, human beings have a negative effect on the impact of natural resource allocation and landscape pattern changes, resulting in the landscape pattern in turn restricting and affecting the development of human society. From 2015 to 2018, the lake area increased by 14.57 km$^2$, mainly due to the treatment projects such as the Yilong Lake retreating and returning to the lake, ecological restoration, and water replenishment of the Yilong Lake since 2013. To some extent, it shows that policy factors have an important impact on land-use change [35].

### 4.2. Driving Forces of Landscape Evolution and Suggestions

Research on the driving mechanism of land use/cover change worldwide shows that 60% of land use/cover change can be attributed to human economic and social development activities, and 40% of land use/cover change is related to factors such as climate change [36]. However, our research shows that the land use in the Yilong Lake Basin has changed dramatically in the past 30 years, which is mainly reflected in the impact of human economic and social development activities. Due to the development of economic construction and the improvement of living standards, the landscape area and the number of patches have changed dramatically, and the landscape pattern index has also fluctuated significantly. On the other hand, the conversion of cultivated land into construction land due to urban

development and construction, road construction and other activities has also exacerbated the fragmentation of the landscape pattern [37]. Human activities are the main driving factor leading to the change inland-type distribution area and spatial pattern in the Yilong Lake Basin, which is similar to the research results in other regions [38,39]. The expansion of construction land area is usually at the cost of the reduction in cultivated land and forest area. The greater the number of patches, the greater the patch density, which means the higher the landscape fragmentation [40]. The patch density index of all land-use types showed an upward trend in the past 30 years, and the patch density was significantly positively correlated with the population, the non-rural population, and the value of primary, secondary, and tertiary industries. The change in transportation land is the most dramatic. From 1990 to 2018, construction land increased rapidly, and the fastest period of area increase was from2000 to 2015. With the growth of population and the expansion of urbanization, the dramatic changes in landscape types have changed the structure and function of the ecosystem, showing that the construction land is mainly far away from the urban area for expansion and development. Therefore, it is suggested to adopt effective ecological restoration for landscape types with broken landscape and degraded functions to maintain landscape integrity and connectivity, restore ecosystem service functions and values of the basin, and maintain landscape ecological security of the basin.

Natural factors are also the driving factors affecting land-use change. Changes in temperature and precipitation have an impact on land-use types, distribution areas and landscape diversity in the whole basin, but the degree of interpretation is relatively small. Shiping is located in a low-latitude plateau, with a typical monsoon climate and remarkable three-dimensional climate. Under the future climate conditions, the ecosystem of Yilong Lake Basin is at risk of degradation. After ecological water replenishment, the mutual feeding relationship between landscape pattern and driving factors will also be adjusted. Natural factors usually need to affect the land type and landscape by changing the wetland hydrological process, affecting soil nutrients, changing the growth and development of wetland plants, as well as the composition of wetland plant communities, interspecific relationships, etc., and the process has obvious hysteresis [41]. Generally, temperature and precipitation changes can affect the wetland area, type, and spatial distribution pattern of the land through direct and indirect effects on the growth of plants, interspecific relationships, soil nutrient conditions, wetland hydrology, etc., in the basin. The influence of natural factors is mainly on large time scales and spatial scales, with characteristics such as large influence range, long duration, time lag, etc. [42]. Relevant studies have shown that climate change is likely to be the main reason for affecting the water cycle and indirectly affecting water resources [43]. This study also found that the annual precipitation had no significant correlation with the area of all landscape types and the landscape pattern index, and the average temperature was only significant with some indicators, indicating that the correlation between climatic environmental factors and watershed landscape change was not obvious. In addition, the strong interference of human activities makes the watershed landscape evolution mechanism very complex [44]. Therefore, on a small time and spatial scale, compared with the interference of human activities, the contribution of changes in natural factors to the watershed ecosystem is relatively small. It is necessary to analyze and study the process and driving mechanism of wetland landscape change under the combination of multiple factors from a spatial perspective.

## 5. Conclusions

(1) The land use of the Yilong Lake Basin is mainly forest, garden and cultivated land. From 1990 to 2018, the area of construction land continued to increase, and the construction land in the basin was mainly transferred from cultivated land and forests. The disturbance of human activities and the change in natural factors jointly affect the land-use type, spatial distribution and landscape diversity of the Yilong Lake Basin, but their contribution to the ecosystem of the basin is different. The explanation degree of human disturbance to wetland area and landscape diversity index is 80.97%, and the

explanation degree of natural factor change is 13.90%. Economic growth, population growth, social affluence and agricultural development are the main socio-economic driving factors of land-use change in the Yilong Lake Basin, and the driving effects of driving factors on land-use change in the basin are increasing year by year. According to the correlation analysis, the driving effects of socio-economic driving factors on land-use change are different. Among them, economic growth, especially industrial structure adjustment, agricultural development and residents' living standards, are the main driving forces affecting cultivated land and forests; population growth and residents' living standards are the main driving forces of the waters. The main factors affecting construction land are population growth, agricultural development, and economic growth.

(2) With the social and economic development, the land-use structure of the basin has been continuously optimized, and the degree of land-use intensification has gradually increased. The shape of landscape patches in the basin is complicated, and the landscape fragmentation and diversity are increased. Urban land and rural residential areas show continuous growth, whereas cultivated land shows a trend of fragmentation. The leap-forward growth of urban land is the main cause of the complexity of the shape of watershed landscape patches. With the ecological restoration of the Yilong Lake Basin, the implementation of the ecological protection and governance of the Yilong Lake surface mountain, the return of the pond to the lake and the ecological restoration, the replenishment and water supply of Yilong Lake and other governance projects, the study on the change in the landscape pattern index shows that the number and density of patches are decreasing as a whole, which means that the landscape fragmentation is smaller, making the landscape types richer, indicating that the important landscape types in the Yilong Lake Basin are protected, and the ecological environment is gradually improving. Therefore, reducing the excessive impact of human activities on the watershed ecosystem is the key to realize the ecological protection of the Yilong Lake Basin.

**Author Contributions:** Writing draft, G.M. and R.Z.; Data curation, G.M. and Q.L.; writing—review and editing, G.S. and L.Z.; methodology, G.M., S.Y. and J.X.; Supervision, G.S. and L.Z. All authors have read and agreed to the published version of the manuscript.

**Funding:** This research was supported by Ecological security assessment of Yunnan Plateau Lake Basin—a case study of Fuxian Lake Basin (Grant No.2022018).

**Institutional Review Board Statement:** Not applicable.

**Informed Consent Statement:** Not applicable.

**Data Availability Statement:** Not applicable.

**Acknowledgments:** We are very grateful to the researchers of Yilong Lake Administration for their data support and help in field surveys. We sincerely acknowledge the constructive comments of anonymous reviewers.

**Conflicts of Interest:** The authors declare no conflict of interest.

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
