# Peer review of "Analysis of Landscape Pattern Evolution and Driving Forces Based on Land-Use Changes: A Case Study of Yilong Lake Watershed on Yunnan-Guizhou Plateau"

_land, doi:10.3390/land11081276_

Round 1
Author Response
Comments of reviewer 1:
- Research field
- Please enlarge the map with the location of the study area.
- Please provide more detailed information about the morphological characteristics and the area of the study area (one sentence is enough)
Revision:
- Thank you for your comments. According to the modification suggestions put forward by experts, the study area map has been enlarged and the pictures in the following text have been adjusted accordingly.
- Thank you for your comments. The location significance and drainage area of the research area have been supplemented
- Method
- In this part of the paper, there are quite a lot of abbreviations, which may be difficult to understand. That is why I suggest inserting a table (as an appendix) to list the meanings of these abbreviations.
Revision:
- Thank you for your comments. The data structure of this paper has been readjusted, and the land use type has been reduced. Therefore, explanatory abbreviations have not been used in the new table, and the abbreviations appearing in the text have been changed to the full name.
- Results
- Use hm2 as the surface measurement unit. Do you mean 100 meters or square kilometers? If it is 100 meters, I suggest converting it into hectares or square kilometers;
- Figure 3 must be enlarged. In these sizes, the change is invisible, so the image does not provide useful information.
Revision:
- Thank you for your comments. The surface measurement unit of hm2 used in the full text has been changed to km2
- Thank you for your comments. Based on the readjusted land use type, figure 3 is changed to figure 2, and the picture is adjusted accordingly.
- Discussion
- Try to compare your results with several results in the international literature. I see that your bibliography focuses more on the papers in nearby areas. Because the magazine is international, your results should be seen in a broader context. I have brought some papers related to your topic to help you.
Revision:
- Thank you for your comments. I have read these articles you gave me. In view of the limitations of the scope of this study area, it can only represent similar ecological areas in adjacent areas, and can not meet the conditions of extension discussion and international, so it can only refer to the relevant theories in the literature.
- Conclusion
- Please fill in your future research methods based on the current results.
Revision:
- Thank you for your comments. According to the current adjusted paper data and the corresponding modified results, the conclusions have been reorganized and the prospects have been prospected.

Reviewer 2 Report
This paper is presented as an interesting and current study, as the subject of human footprint is approached more and more often, especially in the context of population growth and the impact regarding LULC changes over wetlands areas.
I appreciate the approach of the subject, which combines mathematical models with methods of spatial representation, which facilitates the understanding of the discussed phenomenon.
However, I believe that for a better understanding of this work some modifications are necessary, which I will highlight below. My analysis was done by chapters, but my remarks will only focus on part of them.
Introduction: This chapter respects the scientific character of such a paper, the general topic of the article and of the study area being outlined and explained by a sufficient number of quotations (Citations).
Materials and methods:
Study area:
· Please enlarge the map with the location of the study area.
· Please provide more details about the morphometric characters and the area of the study area (one sentence is enough)
Methods:
· In this part of the paper there are quite a lot of acronyms and it can be hard to follow, which is why I recommend inserting a table (as an appendix) with the meaning of these abbreviations.
Results:
· Use hm2 as a surface measurement unit. Do you mean hectometres, or did you mean km2? If it is hectometres, I recommend converting it to either hectares or km2;
· Figure 3 must be enlarged. At these sizes, the changes are not visible, so the image does not present usable information.
Discussions:
· Try to compare your results with several results in the international literature. I see that your bibliography focuses more on papers from the proximal region. Since the journal is an international one, your results should be seen in a much broader context. I come to your aid with a number of papers that lend themselves to your topic.
· Examples from several regions of the globe:
· https://link.springer.com/chapter/10.1007/978-3-540-78648-1_19
https://www.mdpi.com/2072-4292/12/13/2075
· https://www.mdpi.com/2073-4441/10/1/9
· https://www.sciencedirect.com/science/article/abs/pii/S0167880901001815
· https://link.springer.com/article/10.1007/s11269-014-0749-1
· https://link.springer.com/article/10.1007/s00267-014-0332-9
· https://www.mdpi.com/2073-445X/11/5/672
· https://link.springer.com/article/10.1023/A:1006486607040
Conclusions:
· please fill in what would be your future research approaches, based on the current results.
Author Response
Comments of reviewer 2:
- I am looking forward to reading this paper, but unfortunately it is difficult to understand. This paper should not be published in its current form. I suggest the author revise the language and grammar thoroughly. English needs review. There are several strange sentences with grammatical errors. Some sentences are meaningless.
Revision:
- Thank you for your comments. The language and grammar in the paper have been combed according to the readjusted data and structure of the paper, and the content of the full text has been modified accordingly.
- This paper also lacks a detailed literature review. Why did you use these methods in this study? Are there any discussions about these methods in other literatures? Discuss these here. What are some popular methods used to study the evolution of landscape pattern? What is the significance of using land use dynamics, land use transfer matrix or landscape index analysis? The main purpose of this paper is not clear. For all these reasons, I do not recommend this paper in its current form.
Revision:
- Thank you for your comments. Rewrite the full text according to the adjusted paper data and structure. In this study, a high-precision spatial distribution information database of Yilong Lake Basin has been established based on 7 phases of Landsat5TM/8OLI remote sensing image data sources from 1990 to 2018 by combining supervised classification and manual visual interpretation, and 7 land use types have been extracted. This paper studies the change characteristics of land use types, distribution and spatial pattern in the Yilong Lake Basin in recent 30 years, and discusses the relationship between land use types and the interference of local social and economic development and natural factors based on correlation and principal component analysis. Mentioned in the preface. The significance of using land use dynamic degree, land use transfer matrix or landscape index analysis is to study the dynamic changes of landscape pattern in Yilong Lake Basin in recent 30 years, as well as the interference of local social and economic development and the relationship between natural factors.

Reviewer 3 Report
I was looking forward to reading the paper but unfortunately this paper was a tough read. The paper should not be published in its current form. I would advise the authors to thoroughly revise the language and grammar. English language needs to be reviewed. There were several odd sentences with bad grammar. Some sentences just don't make sense.
The paper is also lacking a detailed literature review. Why have you used the methods that you have in this study? Is there any discussion of these methods in other literature? Discuss those here. What are some of the prevalent methods used to examine evolution of landscape pattern? What is the significance of using dynamic degree of land use, land use transfer matrix, or landscape index analysis? The main objective of this paper is not clear. For all these reasons I do not recommend this paper in its current form.
Author Response
Comments of reviewer 3:
- Abstract
- The whole abstract does not present the paper as a whole, but only repeats the major results.
- The first two sentences seem incomplete and used in internal discussion.
- What are "comprehensive land use types"?
- several abbreviations are used already in the abstract without explanation.
Revision:
- Thank you for your comments. This paper has adjusted the data and modified the content of the text. The summary has been reorganized according to the modified content, and the "comprehensive land use type" has been explained. The whole process of using index terms has also been corrected in the text.
- Introduction
- The Introduction in general is quite superficial and short and should be extended.
- References should be improved, several facts and statements are not covered to a full degree. Materials and Methods
- please write also the EPSG Code of the GCS
- in 2.2: What exactly was done in during the "spatial geographic correction"?
- in 2.2: What is meant with "comprehensive analysys" to establish remote sension interpretation marks?
Revision:
- Thank you for your comments. The introduction part reorganizes the logical text according to the references, and makes a certain expansion; Materials and methods have also been reflected in the introduction.
- Thank you for your comments. Method section to add the coordinate system of the image. RGB band synthesis and spatial geographic correction are carried out on the image.
- Thank you for your comments. The spatial resolution of the image data is 30 m. based on the comprehensive analysis mark, it is to better obtain the land use type data of the study area.
- Results
- It would be good to see all the parameters, mentioned in the M&M as plots and graphs, e.g. the dynamic degree. 1
- Table 1 and Figure 3/4/5 are barely readable and need to be improved. E.g. not only land use codes but also the land use name, better formatting.
- The values of the landscape pattern indices need explanation. What exactly means a value of 7 or 20 or 100?
- Figure 6 is not described in the text.
Revision:
- Thank you for your comments. In view of the problem that the parameters mentioned by experts are regarded as drawing and graphics, based on the consideration of the full text, there are too many parameters when drawing pictures, which leads to poor picture display effect. For the problem that the parameters in Table 1 and figure 3/4/5 are almost unreadable, it is necessary to modify the name of land use type in Table 1. If the parameters in figure 3/4/5 are too large, the code has also been marked in the method.
- Thank you for your comments. The abbreviation of landscape pattern index in the text has been changed to the whole process, and the unit symbol has been marked in the table. Figure 6 has been deleted and is only based on principal component and correlation analysis.
- Discussion
- Some sentences are difficult to understand: — "mainly because the Yilong lake has been retreated from the pond to the lake" — "The dynamic degree of comprehensive land use type change in the Yilong lake watershed is gradually increasing"
Revision:
- Thank you for your comments. The discussion part of this paper has been reorganized according to the results of the adjusted data.
- Other comments
There are some typos and missing words throughout the text, please correct.
Revision:
- Thank you for your comments. The full text is reorganized according to the adjusted data, and the spelling errors and missing words are checked in the full text.
Thank you very much for the valuable comments made by the three reviewers!

Round 2
Reviewer 3 Report
The authors have addressed my comments. I recommend publishing the paper.